# Molecular Insight into Iron Homeostasis of Acute Myeloid Leukemia Blasts

**DOI:** 10.3390/ijms241814307

**Published:** 2023-09-19

**Authors:** Emmanuel Pourcelot, Ghina El Samra, Pascal Mossuz, Jean-Marc Moulis

**Affiliations:** 1Laboratory of Fundamental and Applied Bioenergetics (LBFA), University Grenoble Alpes, INSERM U1055, 38000 Grenoble, France; pourcelot.emmanuel@gmail.com (E.P.); ghina_elsamra@outlook.com (G.E.S.); 2Department of Biological Hematology, Institute of Biology and Pathology, Hospital of Grenoble Alpes (CHUGA), CS 20217, 38043 Grenoble, CEDEX a9, France; pmossuz@chu-grenoble.fr; 3Team “Epigenetic and Cellular Signaling”, Institute for Advanced Biosciences, University Grenoble Alpes (UGA), INSERM U1209/CNRS 5309, 38700 Grenoble, France; 4University Grenoble Alpes, CEA, IRIG, 38000 Grenoble, France

**Keywords:** iron regulatory proteins, transferrin receptor, ferritin, proliferation of blasts, ferroptosis, hierarchical clustering, leukemia

## Abstract

Acute myeloid leukemia (AML) remains a disease of gloomy prognosis despite intense efforts to understand its molecular foundations and to find efficient treatments. In search of new characteristic features of AML blasts, we first examined experimental conditions supporting the amplification of hematological CD34^+^ progenitors ex vivo. Both AML blasts and healthy progenitors heavily depended on iron availability. However, even if known features, such as easier engagement in the cell cycle and amplification factor by healthy progenitors, were observed, multiplying progenitors in a fully defined medium is not readily obtained without modifying their cellular characteristics. As such, we measured selected molecular data including mRNA, proteins, and activities right after isolation. Leukemic blasts showed clear signs of metabolic and signaling shifts as already known, and we provide unprecedented data emphasizing disturbed cellular iron homeostasis in these blasts. The combined quantitative data relative to the latter pathway allowed us to stratify the studied patients in two sets with different iron status. This categorization is likely to impact the efficiency of several therapeutic strategies targeting cellular iron handling that may be applied to eradicate AML blasts.

## 1. Introduction

Within the large category of leukemia, acute myeloid leukemia (AML) is characterized by the active development of neoplastic cells (blasts) in the bone marrow environment of hematopoietic stem cells and progenitors, and they ultimate release in the blood stream. The differential program of AML cells can be blocked at different stages of hematopoiesis, and the morphological and cytochemical traits of the leukemic cells were initially used to classify them (FAB classification) [1]. However, with the exception of acute promyelocytic leukemia which presents with the t(15;17)(q24;q21) translocation involving the gene of retinoic acid receptor α, the small number of first-defined AML sub-types hides a very large combination of different mutations and a wealth of mechanisms for maintenance and growth of leukemic blasts. Consequently, the FAB classification has been overridden by molecular sub-sets, and the therapeutic regimens have expanded to take the diversity of the disease into account [2]. However, recent evaluations of 5-year survival after diagnosis for US adults remains lower than 30% [3], which prompts the search for new paradigms to more efficiently treat this deadly disease.

Generally speaking, intensive chemotherapy and autologous or allogeneic stem cell transplantation are the treatments of choice for AML. But for patients who cannot endure such strong treatments, chemotherapy must be adjusted to their condition. Leukemic cells are intrinsically heterogeneous at the genetic, metabolic, signaling, and other levels [2,4,5]. Therefore, beyond the identification of the oncogenic mutations present in patients’ cells, it appears useful to obtain a more in-depth mechanistic knowledge of these cells and their growth modalities to design informed and better therapeutic strategies.

Erythropoiesis refers to the production of red blood cells, and it is the largest iron-consuming pathway in metazoan biology for the synthesis of hemoglobin [6]. However, iron has many more qualitatively important biological roles in different cells [7], including hematopoietic ones [8,9]. As such, and considering the growth requirements of cancerous cells, the links between iron and oncogenesis have been heavily documented [10]. Consequently, manipulating the iron homeostasis of cancer cells can be an efficient strategy to selectively remove them [11,12]. However, such approaches are more likely to succeed if the detailed iron status of the targeted cells is precisely known. 

As such, we explored iron-related aspects of the ability of healthy CD34^+^ cord blood progenitors and AML blast cells to proliferate in a synthetic and perfectly defined medium. Immediately after isolation, several molecular traits distinguished leukemic blasts and healthy progenitors. Selected variables of cellular iron homeostasis allowed us to separate leukemic blasts samples in two groups: this discrimination has consequences for the internal wiring of these cells and for the efficiency of planned or implemented therapeutic strategies.

## 2. Results

### 2.1. Proliferation Status of Cord Blood Progenitors and of Leukemic Blasts

When isolated, cord blood progenitors defined as CD34^+^ cells were quiescent and blocked in the G0/G1 phase of the cell cycle (Figure 1A), with cells in G2/M representing less than 1% of the total population. Attempts at amplifying cord blood CD34^+^ cells either in a commercial medium of not completely available composition, but which contained 2.3 µM iron as measured by inductively coupled plasma mass spectrometry, or in a home-made synthetic minimal medium (SMM-1) for comparative purposes, resulted in a transient increase 6 days after a medium change in the former, and a regular increase in the latter over 10 days after several medium changes (Appendix A). However, the proportion of CD34^+^CD38^+^ cells (Appendix A), as well as those of the global CD34^+^ fraction and of the CD34^+^CD38^−^ one, decreased in the amplified populations after 3 days and a medium change for SMM-1, and 5 days in the commercial medium. In parallel, the cytological morphology of the cells became mainly that of (pro-)monoblasts as soon as 6 days after seeding. Amplification of cord blood CD34^+^ progenitors was clearly accompanied by entrance into the cell cycle (Figure 1B). 

Leukemic cells (blasts) from the bone marrow of AML patients were very weakly proliferative when isolated, as exemplified in Figure 1C, irrespective of their immuno-phenotype (CD 34 positive or negative). Deconvolution of the cell cycle histograms of all analyzed samples with the Modfit program gave a mean of 93.3% of cells at the G0-G1/S checkpoint phases (SD 4.7; *n* = 7). A very sensitive marker of dividing cells is ribonucleotide reductase, which synthesizes deoxyribonucleotides, the DNA building blocks, from ribonucleotides. It is absent in quiescent cells [13,14,15]. Therefore, the levels of the R2 subunit transcripts were compared between healthy CD34^+^ cells and leukemic blasts. Cumulative data (Figure 1D) indicated that whereas the RRM2 transcript of healthy CD34^+^ cells was low (mean value 0.00269 SD 0.00244 relative to normalizers), it was larger, although scattered and not abundant either, for leukemic blasts (mean 0.0488 SD 0.0723). These data support the idea of a larger proliferation-prone fraction of cells in patients’ samples as compared to non-leukemic CD34^+^ cells, since a *t*-test between the two groups indicated significance (*p* < 0.001). 

Attempts at amplifying leukemic blasts as above with healthy progenitors failed to increase the small proportion of cells in the S and G2/M phases and rather resulted in a large amount of dying cells with fragmented DNA (Figure 1E).

### 2.2. Iron Dependence of Amplification of Cord Blood Progenitors and Leukemic Blasts

When the cord blood progenitors were seeded in the SMM-1 medium without iron provided by transferrin (Tf), viability was maintained for 3 days but cells hardly expanded (Figure 2A, leftmost bar). With the addition of iron from 1.25 µM fully-saturated Tf, proliferation occurred, and the number of cells clearly increased (Figure 2A). The mean amplification of 9 batches of different cord blood donors was 1.12 (SD 0.306) without added Tf, and that of 12 different batches was 2.61 (SD 0.75) with 1.25 µM Tf. For leukemic blasts, the respective values were 0.59 (SD 0.12) and 0.92 (SD 0.02). A *t*-test run on the increased amplification factors with Tf for healthy progenitors and AML blasts indicated a significant difference between the two (*p* = 0.002). Therefore, iron provision is needed to maintain and amplify cord blood progenitors over the time scale (3 days) during which they do not commit to the early steps of differentiation (Appendix A). In more detail, amplification of cord blood CD34^+^ cells over 3 days was dependent on the amount of Tf, (hence, of iron) provided in the growth medium (Figure 2B). The number of cells rapidly increased with the Tf concentration and plateaued above ca. 100 nM. One-way ANOVA indicated significant differences in the mean amplification values (*p* = 0.001), and pairwise comparisons distinguished data at Tf concentrations above 200 nM from those without added Tf (*p* < 0.05). Reciprocally, the amount of transferrin receptor was inversely proportional to the transferrin concentration added to the growth medium (Figure 2C,D). Whereas this value was relatively low for cord blood CD34^+^ cells immediately after purification (leftmost lane of Figure 2C) as expected for nearly quiescent cells (Figure 1A), it increased after 3 days in the SMM-1 medium without added Tf (Figure 2C). With added Tf, cells were progressively replete in iron, and the transferrin receptor levels returned to the value of the as isolated cord blood CD34^+^ cells above ca. 20 nM Tf (Figure 2D).

In contrast, assays examining the preservation and amplification of leukemic cells in the same conditions failed to give similar results to cord blood cells. The leukemic blasts lost viability without added Tf, with only ca. 20% of the initial cells remaining 5 days after seeding. The addition of saturating Tf iron significantly limited the loss of viability of leukemic blasts, but the blast population still decreased between seeding and the 3 day ex vivo culture (Figure 2A). These results apply to cells of primary AML as well as to cells of the AML transformation from myelodysplastic syndromes (secondary AML), even though secondary AML patients often receive regular transfusions, and their blasts display more cells in the S-phase of the cycle at harvest (ca. 35%) than observed with primary AML (*ca*. 6.5% Figure 1C). This difference did not change the ability of blasts to grow or survive in SMM-1.

Therefore, healthy progenitors and leukemic blasts cannot be easily and efficiently maintained or amplified in a thoroughly controlled liquid culture without significant cellular modifications. As such, characterizing the differences between healthy immature hematopoietic cells and leukemic blasts requires us to study them right after isolation. 

### 2.3. Is the Difference in Survival between AML Blasts and Healthy CD34^+^ Progenitors on Transferrin-Provided Iron Due to Different Transferrin Receptor Levels?

A good deal of the following results rely on measurements of mRNA from genes of interest. As such, we estimated the variability of expression of some genes among a series of healthy samples obtained from cord blood samples, bone marrow aspirates of adult patients receiving cardiac surgery, and granulocyte colony-stimulating factor-mobilized CD34^+^ cells from young adults’ bone marrow. As shown in Appendix A, the coefficient of variation is of the order of 50% or below for the most expressed genes on the list, and it is moderately (+10%) larger for the less expressed ones. Hence, differences in expression beyond CV% exceed individual variability.

Iron is a prominent component of molecular systems that provide energy to cells, with specific involvement in proliferation and growth of cancer cells [12]. Pyruvate dehydrogenase kinases, PDKs, repress the conversion of pyruvate to acetyl-CoA and the fueling of the tricarboxylic acids cycle [16]. When comparing the levels of PDK1 relative to three normalizers, expression in patients was larger than that of non-leukemic CD34^+^ cells (Appendix A). However, the PDK1 protein was not significantly (*p* = 0.093) more abundant in the patients’ samples than in non-leukemic bone marrow CD34^+^. When pyruvate is diverted from the tricarboxylic acids cycle, it is reduced to lactate by NADH. The levels of expression of lactate dehydrogenase A (LDHA) were higher in AML than in healthy cells (Appendix A). Therefore, both PDK1 and LDHA are overexpressed on average in AML cells. This might be taken as an indication for a shift of catabolism from oxidative phosphorylation to glycolysis in the studied samples, as is often observed in cancer cells [17].

The transferrin–transferrin receptor endocytosis pathway is the principal entry of iron into hematopoietic cells at many maturation stages [18]. With the available samples, expression of the transferrin receptor gene was larger in AML cells than in non-leukemic ones. A *t*-test (Mann–Whitney rank sum test) between the two groups indicated a significant difference (*p* < 0.001) between the median values (Figure 2E). Thus, the apparently stronger proliferation rates of AML blasts populations compared to that of non-leukemic progenitors (Figure 1D) are paralleled by larger TFRC expression levels in AML cells. Of note, only approximately one-fourth of the AML blasts samples contributed significantly (*p* < 0.014, ANOVA on ranks/sample, samples compared to controls by Dunn’s test) to the higher levels of TFRC mRNA compared to non-leukemic CD34^+^ cells. The other AML blast samples were not different from the healthy CD34^+^ cells in this respect, but none had less TFRC mRNA. In contrast to mRNA, measurement of the transferrin receptor protein failed to indicate significant differences between AML and non-leukemic cells. However, differences among the mean values were consistently observed and have been used as complementary information in analyzing the patients’ samples (see Section 2.6).

### 2.4. The Iron Storage Capacity of Blasts and Consequences on Iron-Induced Stress

From the above observation of global TFRC upregulation in leukemic blasts, it was worth examining whether it was correlated with some increased iron storage capacity involving ferritin. The nearly 4-fold higher median value of ferritin L (FTL) mRNA (Figure 3A) was paralleled by the 2.15-fold larger one for ferritin H (FTH1) mRNA (Figure 3B). The differences among the patients’ samples were very large, by more than two orders of magnitude in some cases. Compared to the pooled control samples, the individual AML samples evidencing significant differences in the ANOVA test for FTH1 (*p* < 0.005) represented a minority. In the case of FTL, the situation was qualitatively similar. Thus, again, as in the case of TFRC, a minority of samples contributes to the statistical difference between the group of controls and that of leukemic blasts. On average, FTH1 expression was higher than that of FTL, by a factor of ca. three or four. 

FTH1, besides its role in iron homeostasis, is also an acute phase protein [19]. Iron imbalance and FTH1 induction may witness redox insult; hence, the expression of the inducible heme oxygenase gene (HMOX1) in AML blasts and CD34^+^ healthy cord blood cells was measured. First, expression of HMOX1 was not or hardly detected in the preparations of most CD34^+^ healthy cord blood progenitors from four different donors. In contrast, most leukemic blasts had variable but often relatively large concentrations of HMOX1 mRNA (Figure 3C). When referred to the expression level of the constitutive heme oxygenase, HMOX2, the HMOX1/HMOX2 ratio was larger in leukemic blasts than in cord blood cells (Figure 3D). The HMOX1 transcripts varied over a 50-fold larger scale than that of HMOX2. Some of the samples displayed values close to or smaller than control cord blood samples, whereas others were far larger (note the log y-scale in Figure 3C). 

### 2.5. The Post-Transcriptional Regulation of Iron in AML Blasts

The relative levels of the ferritin subunit encoded by FTH1 in 22 samples did not display strong differences between leukemic and non-leukemic samples, as in the case of the transferrin receptor (see Section 2.3). Furthermore, the absence of a clear positive regression between the levels of the FTH1 mRNA and of the encoded protein (Appendix A) suggested the involvement of post-transcriptional regulation. 

At the cellular level, iron homeostasis, and ferritin translation in particular, are dependent on the activity of the iron regulatory proteins. Although the most documented regulatory step of the iron regulatory proteins is post-transcriptional, the genes may be differentially expressed depending on the environmental conditions [20,21,22]. However, the mRNA levels of both ACO1 (IRP1) and IREB2 (IRP2) genes in AML blasts and in non-leukemic CD34^+^ cells (Appendix A) were not statistically different (*t*-test *p* = 0.132 and 0.198 for ACO1 and IREB2, respectively). 

Regarding proteins, IRP1 mainly balances between two mutually exclusive activities: one in which a [4Fe-4S] cluster is bound and provides the cytosolic aconitase activity of the cells, and the other in which the de-metalated form interacts with the iron responsive element (IRE) of the regulated mRNA [23]. In the case of IRP2, the cellular iron availability signals degradation of the protein, which is, thus, only active under conditions of iron deficiency, or other conditions in which the FBXL5-dependent degradation system is inhibited [24]. FBXL5 is responsive to the assembly of a [2Fe-2S] cluster [25] and leads IRP2 to the proteasome. RNA (i.e., labeled IRE) shift assays may distinguish IRP1 and IRP2. The recombinant pure proteins, that have the exact sequence of the native proteins and are not post-translationally modified [26,27], have been used as references (outmost lanes of Figure 4A). Clearly, lysates of primary AML blasts only display IRP1 activity (Figure 4A). This observation is congruent with that made with some hematopoietic cell lines [28]. Since IRP2 mRNA is produced at roughly the same level as that of IRP1 (Appendix A), the absence of IRP2 activity in leukemic blasts indicates that any synthesized IRP2 is efficiently degraded. The IRP(1) activity of AML blasts, excluding secondary AML, is approximately three times larger than that of non-leukemic CD34^+^ cells (Welch’s *t*-test *p* = 0.012) (Figure 4B), which agrees with the increased TFRC mRNA (Figure 2E) that is stabilized by IRP activity. 

### 2.6. Partitioning of the Leukemic Samples According to Their Iron Status

For those samples for which extensive data were available, partitioning was implemented with the *pam* and *pamk* functions in R using 6 variables related to iron homeostasis, namely the transferrin receptor and ferritin proteins and mRNA, and the IRP activity. These data could be more convincingly split in two partitions. A parallel hierarchical clustering analysis with *diana* produced the pattern shown in Figure 4C, in which the two partitions defined by *pam* (k = 2) are clearly separated. Close examination showed that the IRP activity and the protein concentration of ferritin H were significantly different between the two partitions (Welch’s two-tailed tests, *p* values of 0.00115 and 0.00509, respectively), as represented in Figure 4D,E. These differences indicated that samples in Partition 2 had a lower IRP activity and a larger ferritin H protein concentration than the samples in Partition 1. This inverse relationship between these two variables agrees with the known effect of IRP activity, which represses ferritin translation. In the meantime, the other variables included in the clustering analysis failed to show significant differences between the two sets of partitioned samples.

## 3. Discussion

The detailed characterization of hematopoietic progenitors and of leukemic blasts has been greatly impaired by the difficulties in maintaining and amplifying them in their original status in a perfectly defined fully synthetic medium. Indeed, it is now well established that the detailed conditions in which pluripotent hematopoietic cells are set ex vivo, including nutrients and metabolites, modulates their fate, be it reprogramming, death, or differentiation [29]. Such difficult-to-predict outcomes stem from regulation and combinations of a wealth of transcription, epigenetics, and other signaling pathways. Our presently reported experiments (Figure 1 and Figure 2) confirm these trends. Therefore, we opted for working with as-isolated cells to avoid introducing bias in our analysis that is focused on the unaltered molecular patterns of healthy progenitors and of AML blasts.

Iron provision via transferrin has a significant contribution to the behavior of both non-leukemic CD34^+^ cells and AML blasts ex vivo, with the former being more resistant to cell death than the latter when enough Tf is provided (Figure 2). Most AML patients of the studied set, males and females combined, were anemic at diagnosis, with a mean hemoglobin value of 91.2 g/L (SD 18.7), well below the currently 120 g/L cut-off for anemia relative to non-pregnant adult women (higher for men) defined by the World Health Organization [30]. However, this feature is more easily linked to disease-related inefficient erythropoiesis than to deficient iron homeostasis. Therefore, the precise modulation of iron use in AML blasts as compared to healthy progenitors remains unclear and was worth investigating.

As previously observed, both AML blasts and healthy hematopoietic progenitors are not strongly engaged in the cell cycle when isolated. However, whereas non-leukemic CD34^+^ cells look quiescent (Figure 1A), blasts show a small proportion of proliferating cells (Figure 1C) and signs of a possible metabolic shift toward glycolysis (Appendix A). These features have been regularly reported for cancer cells [31] and AML blasts in particular [32]. Additionally, upregulation of HIF1A was detected in AML blasts, but without clear overproduction of the corresponding protein (Appendix A), whereas expression of HIF2A remained low. These data cannot be used to support the increased hypoxic features generally assigned to AML cells [33], but they are in line with the questioned role of hypoxia in the AML microenvironment [34,35], and the variable distribution of hypoxia inducible factors in sub-populations of leukemic cells [36]. 

The AML blasts display molecular features, such as increased expression of the Tf-receptor and ferritin genes (Figure 2E and Figure 3A,B), that strongly suggest increased intracellular iron fluxes compared to healthy progenitors. These features may result in larger iron availability and use, but this is apparently not the case. The intracellular molecular detector for usable iron that is based on the activity of IRP1 shows that the leukemic blasts look more iron-depleted than healthy progenitors (Figure 4B). The term ‘iron-depleted’ refers here to the inability to use iron for biosynthetic needs, not necessarily a lowered concentration; indeed, ferritin upregulation (Figure 3A,B) is not supposed to occur if iron is lacking in leukemic blasts. Therefore, the data rather indicate some dysregulation of iron homeostasis in AML cells, and, taken together, define a distinct iron status in the leukemic cells compared to the non-leukemic ones.

The involvement of iron dysregulation in cancer has been regularly reviewed [6,10]. The case of AML has recently been more specifically addressed [9,37,38]. Iron management, be it enhanced use or shielding, plays a role in most steps of myeloid differentiation [8,9]. The complexity of the inter-relationship between leukemia and iron homeostasis does not afford straightforward therapeutic interventions to selectively remove cancer cells, but they provide various leads to target them. The present data confirm and develop that iron homeostasis is disturbed in AML, and they have been used to classify patients on the basis of the iron status of their blasts at the molecular level. The outcome is that the segregation of patients’ molecular data relative to cellular (blast) iron homeostasis falls into two main sets. Interestingly, although the used algorithms were fed with variables related to both mRNA and proteins, the splitting between the two sets is mainly triggered by the IRP activity and the H ferritin protein concentration (Figure 4D,E). This is in line with more global attempts at profiling AML patients that emphasized the usefulness of proteomic, as compared to transcriptomic, data [39]. However, the complexity of the regulatory steps leading to the iron status of AML blasts is illustrated by the large differences, in the 1:750 range, of the transferrin receptor mRNA concentrations, and the more modest 1:12.5 range for the corresponding proteins. The variety of regulatory steps in AML blasts modulating the transferrin receptor reduces the part played by the IRP activity, in agreement with the recently reported evidence of dissociation between TFRC and transferrin receptor concentrations, even with explicit consideration of IRP activity, in cell lines [40].

The resulting segregation of AML blasts into two sets (Figure 4C) was challenged by the data obtained for other parameters at diagnosis. Among genetic, clinical, blood count, metabolic, ionic, signaling, and other data, the mean platelet volume (MPV) was higher (+11.7%, Welch’s *t*-test *p* = 0.018) in Partition 2. The patients’ MPV was in the upper end of the reference values, slightly above in Partition 2 (mean 11 fl) and slightly within (mean 9.87 fl) in Partition 1. The mean of the platelet count was also increased (+60%), although not significantly (Welch’s *t*-test two-tailed *p*-value = 0.306), in samples of Partition 2. All patients had thrombocytopenia, as is well-documented in AML [2], and the iron specificity of the samples in Partition 2 (Figure 4D,E) will await additional data to assess its connection to thrombopoiesis, as should be considered [41]. In addition, the ratio of total/protein thiols in the plasma was 30% higher in Partition 2 as compared to Partition 1 (Welch’s *t*-test two-tailed *p* = 0.008). However, since other redox-linked parameters, such as glutathione peroxidase or superoxide dismutase activities and glutathione or malondialdehyde concentrations, were similar in both groups, further consideration of this observation should be postponed until further data become available.

In preliminary retrospective analysis, we noticed that ex vivo survival for 3 days of freshly isolated samples classified in Partition 2 was higher than that of samples in Partition 1. Similarly, the percentage of cells engaged in the cell cycle right after isolation seemed larger in Partition 2 as compared to Partition 1. However, no obvious trend was noticed with overall survival of the corresponding patients or sensitivity to chemotherapy when implemented. Therefore, our data are insufficient to convincingly associate the molecular partitioning of Figure 4C with the chemoresistance of leukemic blasts or proliferation propensity. A new prospective study should be designed to answer such potential correlations.

Our study is limited by the small number of patients for whom extensive data have been obtained. This is particularly prejudicial in view of the large AML heterogeneity [2,4,5]. However, our results are sufficient to provide a valuable basis to further investigate AML and its treatment. Iron may promote the growth of leukemic cells, but it is also the catalyst of cell death by ferroptosis. For instance, dihydroartemisinin, the anti-malarial derivative of the natural compound artemisinin, induces ferroptosis in AML cells by ferritin degradation and intracellular increase in reactive iron [42,43,44]. Dihydroartemisinin’s ability to mobilize reactive iron and to kill AML blasts is, thus, predicted to be higher in samples from Partition 2, since lower IRP activity and higher ferritin content should enhance ferroptosis sensitivity.

In conclusion, the genetic and clonal heterogeneity of AML blasts [2,4,5] has called for a contemporary chemotherapeutic approach combining targeted inhibitors and more conventional anti-proliferative drugs to treat patients [45]. Initially, such strategies have heavily relied on genomic information revealing a series of frequently mutated genes that are now at the core of the classification of the sub-types of the disease [2]. However, more integrated and extensive data are likely to improve the identification of targets and the efficiency of therapeutic strategies. In this respect, additional high-throughput or large-scale methods are instrumental in refining the molecular landscape of AML cells, and in predicting the outcome of various considered interventions [5,46,47]. On the basis of the data reported here, we posit that such approaches may also benefit from the inclusion of molecular and functional data dealing with iron homeostasis, not only for managing AML, but likely also for targeting other kinds of cancer.

## 4. Materials and Methods

### 4.1. Purification of CD34^+^ Progenitors

CD34^+^ cells were obtained from non-leukemic donors after Ficoll-Hypaque (Abcy-Eurobio) density gradient separation, and they were isolated by two steps of immunomagnetic separation (Miltenyi Biotec, Paris, France), first on large- and then on medium-size columns. Cord blood procedures were approved by the French Blood Service’s Institutional Review Board. All subjects gave their informed consent for inclusion before they participated in the study. The study was conducted in accordance with the Declaration of Helsinki, and the protocol was approved by the Ethics Committee of CHUGA (Project 38RC13.209). The main characteristics of the AML samples are presented in Appendix A.

### 4.2. Amplification and Proliferation of CD34^+^ Progenitors and AML Blasts

A synthetic minimal medium (SMM-1) was used to control iron supply. It was composed of Iscove’s modified Dulbecco’s medium (IMDM, Life Technologies, Carlsbad, CA, USA), 10 mg/mL (1%) iron-depleted bovine serum albumin (see below), 200 µg/mL bovine pancreas insulin (Sigma I-1882, St. Louis, MO, USA), 0.1 mM β-mercaptoethanol and the StemMACS HSC expansion cocktail (Miltenyi Biotec 130-100-843), containing recombinant human Flt3-Ligand, recombinant human stem cell factor (SCF), and recombinant thrombopoietin (TPO). In culture media, bovine serum albumin (BSA) is a source of iron [48]. As before [48], to remove iron, the BSA powder was dissolved in 10 mM HEPES buffer pH 7.3 containing 5 mM EDTA before dialysis with successive baths of distilled water until it reached an EDTA concentration below 1 nM. Beforehand, distilled water was filtered through Chelex cation exchange resin (BioRad, 3 bld Raymond Poincaré, 92430 Marnes-la-Coquette, France). For comparison, expansion of cord blood CD34^+^ cells was also carried out in the commercial StemMACS HSC Expansion Media XF human (Miltenyi Biotec 130-100-473) supplemented with the same StemMACS HSC expansion cocktail as mentioned above for SMM-1.

Freshly isolated cells were seeded at 1 × 10^5^ cells/mL and grown for three days at 37 °C in 5% CO_2_ in the SMM-1 minimal medium in which the iron source was precisely monitored by varying the fully saturated (holo-) human transferrin (Sigma-Aldrich T0665, St. Louis, MO, USA) concentration. Viable cells were determined by Trypan blue staining in Neubauer slides. 

### 4.3. Flow Cytometric Analysis

For immunophenotyping of isolated progenitor cells and to monitor evolution upon amplification, flow cytometry was performed with a FACS Calibur^TM^ (Beckton Dickinson, 11, rue Aristide Bergès, ZI des Iles, BP4, 38801 Le Pont de Claix Cedex, France) cytometer. The different phases of the cell cycle were observed after propidium iodide labeling as described [40], and the flow cytometry histograms were analyzed with the Modfit LT v3.2 software (Verity Software House, Topsham, ME, USA). Cells were labelled with VioBlue-conjugated anti-CD34 antibody, phycoerythrin (PE)-conjugated anti-CD38 antibody, and a lineage mix of fluorescein isothiocyanate (FITC)-conjugated anti-CD3, CD19, CD56, CD13, CD14, and CD11a antibodies (Miltenyi Biotech). For differentiation assay, a complementary labelling was carried out with Brilliant Violet-conjugated anti-CD34 (Beckton Dickinson Pharmingen), FITC-conjugated anti-CD36 (Beckman Coulter, Villepinte, France), PE-conjugated anti-CD235a (GPA) (Beckman Coulter), and allophycocyanin (APC)-conjugated anti-CD71 (Beckman Coulter) antibodies. 

### 4.4. Western Blotting and RNA Electrophoretic Mobility Shift Assays (REMSA)

The implemented procedures were as recently recalled [40], with the BlotCycler automated western blot apparatus (Precision Biosystems, Mansfield, MA, USA). All blots, Western and REMSA, were developed with the Amersham ECL Prime Western Blotting Detection Reagent (Amersham Biosciences, Cat. No. RNP2232, Amersham Place, Little Chalfont, Buckinghamshire, UK HP7 9NA) and imaged on an ImageQuant LAS4000 system (GE Healthcare Bio-Sciences AB, Björkgatan 30, 751 84 Uppsala, Sweden). In Western blotting experiments, guanidine solutions were used for successive probing of the same polyvinylidene fluoride (PVDF) membrane [49] for soluble proteins, and the samples were normalized for loading differences with the signal of β-actin or tubulin depending on the probed size range. The transferrin receptor protein was detected on nitrocellulose membranes and stained with Ponceau Red for normalization. Appendix A lists the antibodies. Images of Western blots were quantified with the Image J software https://imagej.nih.gov/ij/ (accessed on 21 September 2021). The background was subtracted when needed. 

The RNA electrophoretic mobility shift assays (REMSA) used the human ferritin H 5′-GGUUUCCUGCUUCAA*CAGUGC*UUGGACGGAACCC-3′ (apical loop italicized) iron responsive element. The synthetic probe was 3′ labeled with biotinylated cytidine bisphosphate with T4 RNA ligase (ThermoFischer Scientific, 1200 Bd Sébastien Brant, 67400 Illkirch-Graffenstaden, France). Reaction of the labeled probe with the iron regulatory proteins was separated on cooled non-denaturing 8% polyacrylamide gels run in 0.25× Tris-Borate-EDTA at 13,500 V.min. The separated proteins were transferred onto Hybond N+ (Amersham) membranes, the nucleic acids were cross-linked with a UV lamp (254 nm, 8 W), and the biotinylated probe was detected with the Chemiluminescent Nucleic Acid detection module (ThermoFisher Scientific). The membrane was immediately processed as above for Western blots. The bands of the resulting images (8-bits) were quantified following the procedure outlined at https://lukemiller.org/index.php/2010/11/analyzing-gels-and-western-blots-with-image-j/ (accessed on 11 July 2019), and the relative densities were calculated with reference to that of a known quantity of recombinant IRP1 [26] loaded in one lane of the gel for this purpose. In this way, the absolute amount of active IRP could be calculated relative to the quantity of proteins in cell lysates.

### 4.5. Reverse Transcriptase-Quantitative Polymerase Chain Reaction (RT-qPCR)

RNA purification, reverse transcription, and qPCR amplification, at least in triplicate, were performed as previously reported [28,40]. Normalizers were the RPLP0 (ribosomal protein lateral stalk subunit P0), hypoxanthine-guanine phosphoribosyltransferase (HPRT), and β-actin genes, which were amplified in parallel with the genes of interest. The oligonucleotides used to detect expression of the different genes are listed in Appendix A. To compare the controls’ and patients’ samples, all measurements of each group were pooled, plotted, and the statistical analysis of the comparison was carried out.

### 4.6. Graphical Representation of the Data

Forest plots were used to compare the data relative to AML and non-leukemic cells. The boundaries of the boxes were the 25th and 75th percentiles, with the internal lines marking the median values, and outlying data points were explicitly indicated as black dots. The whiskers were the 90th and 10th percentiles.

### 4.7. Statistical Analysis

Results are reported as means ± standard deviation (SD). Means of two groups were compared with Welch’s *t*-test without assuming equal variances. The two-tailed *p*-values were used to estimate the significance of differences. One way ANOVA followed by Holm–Sidak tests were used for the comparisons between several groups. Partitioning of patients’ data and hierarchical clustering were carried out with algorithms included in the R software (v.4.2.2 https://www.r-project.org/ (accessed on 3 February 2023)). The algorithms *pam* (partitioning around medoids) [50] in the cluster package and *pamk* in the flexible procedures for clustering (v.2.2-10) [51] were implemented. Each variable was scaled by dividing the values by their root-mean-square. The results of partitioning with *pam* and *pamk* were probed by the use of k-means, and no qualitative differences in the results between the various methods were observed. Divisive hierarchical clustering was implemented with *diana* (DIvise ANAlysis), as a way of first identifying large clusters [50]. Comparison with the agglomerative clustering algorithm *agnes* (AGglomerative NESting) did not reveal differences, and both methods converged in the segregation of the data in only two main groups, as described in Section 2.6.

## Figures and Tables

**Figure 1 ijms-24-14307-f001:**
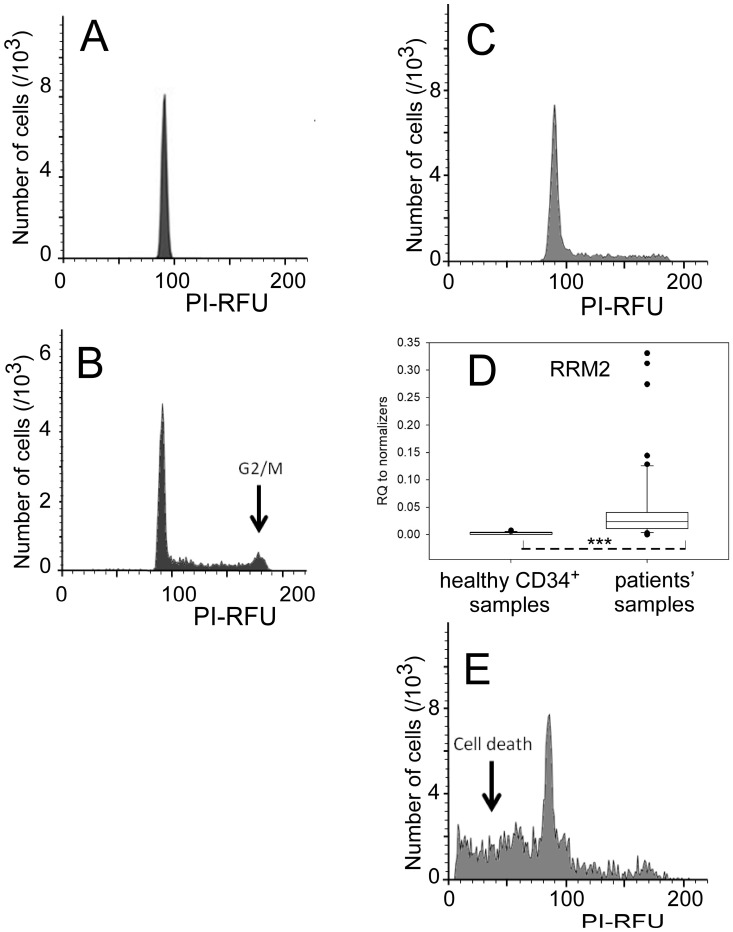
Proliferative status and amplification of cord blood CD34^+^ progenitors and AML blasts in the SMM-1 synthetic minimal medium. (**A**) Distribution of the cell population in freshly isolated cord blood CD34^+^ progenitors. The cellular DNA was labeled with propidium iodide, the fluorescence of which (PI-RFU) is plotted along the x-axis, with the number of thousands of cells along the y-axis. (**B**) Distribution of the cell population after amplification of cord blood CD34^+^ progenitors in SMM-1 for 3 days. The data are plotted as in (**A**). (**C**) Representative histogram of the cell population in a freshly isolated leukemic sample. Bone marrow blasts from one AML patient were analyzed as in (**A**). (**D**) Amounts of RRM2 mRNA relative to the 3 normalizers, RPLP0, HPRT, and ACTB. *** *p* < 0.001. The samples included 5 healthy donors, 3 providing cord blood and 2 bone marrow aspirates, and 20 AML patients. The plot is the result of 23 and 55 measurements for healthy donors and patients, respectively. (**E**) Analysis of the sample of (**C**) after 3 days in SMM-1.

**Figure 2 ijms-24-14307-f002:**
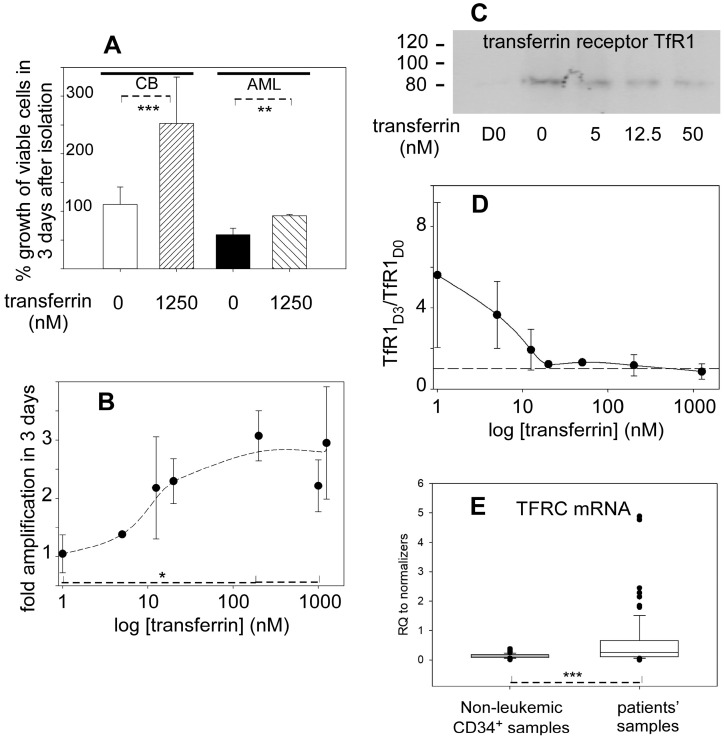
Iron dependence for growth of progenitors and leukemic blasts. (**A**) Transferrin (Tf) enhances amplification of hematopoietic progenitors and of leukemic blasts. The panel shows the variations in viable cells (%) after 3 days in the serum-free medium (SMM-1) with only residual iron contained in the media (0 nM added) or with a large transferrin concentration (1250 nM) for CD34^+^ cord blood cells (left lanes) and leukemic blasts (right lanes). The % label = 100 on the y-axis corresponds to the number of cells seeded at the start of the experiment (day 0). ** *p* < 0.01 *** *p* < 0.001. (**B**) Amplification of cord blood progenitors as a function of iron provision. Isolated cord blood CD34^+^ cells were incubated in the SMM-1 medium with the indicated fully-loaded transferrin concentrations. Cells were counted after 3 days and the measured number was divided by that of the inoculum to give the amplification ratio plotted along the y-axis. The x-axis is logarithmic for clarity, and the samples to which no transferrin was added are given a value of 1. The drawn dashed line is a spline curve through the experimental points that are plotted with SD. This plot is the overlap of experiments carried out with 9 different cord blood batches. * *p* < 0.05. (**C**) Levels of transferrin receptor in cord blood progenitors as a function of transferrin provision. One representative sample used to build panel (**B**) was lysed after 3 days and the transferrin receptor was measured by Western blotting in the particulate fraction. The D0 lane corresponds to the sample obtained right after isolation, and equal loading was verified by Ponceau Red staining of the membrane. The position of molecular markers in kDa is indicated on the left. (**D**) Plot of the relative transferrin receptor intensities with SD obtained with 5 different cord blood CD34^+^ preparations. The x-axis of added transferrin is logarithmic for clarity. The y-axis (TfR1_D3_/TfR1_D0_) is the ratio of the transferrin receptor intensity after 3 days of culture to that right after isolation, arbitrarily set to 1 (dashed line). The solid line is a spline curve through the experimental points. (**E**) Expression of TFRC relative to the 3 normalizers, namely RPLP0, HPRT, and ACTB. The samples included 10 healthy donors, 3 providing cord blood and 7 bone marrow aspirates, and 21 AML patients. The plot is the result of 65 and 85 measurements for non-leukemic donors and patients, respectively. *** *p* < 0.001.

**Figure 3 ijms-24-14307-f003:**
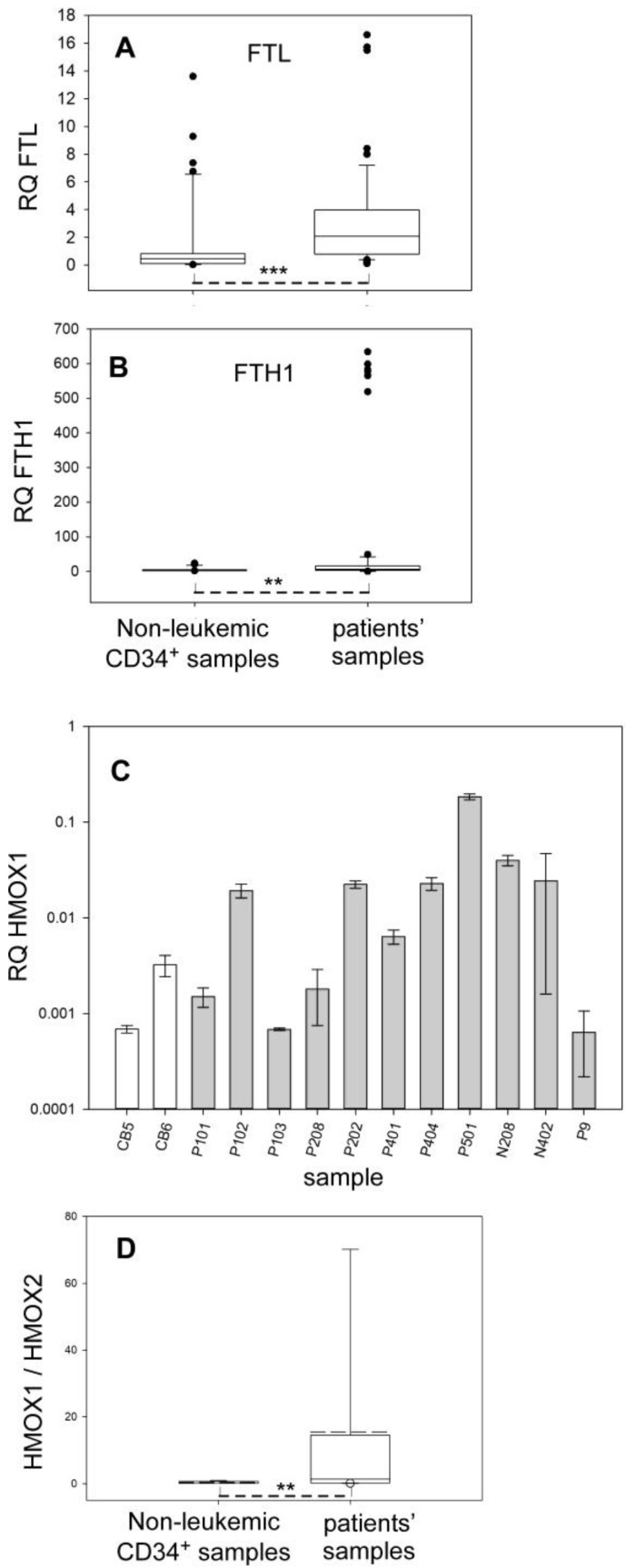
Expression of ferritin and heme oxygenase genes in AML and non-leukemic CD34^+^ cells. The quantity (RQ) of the transcripts is relative to the 3 normalizers, namely RPLP0, HPRT, and ACTB. ** *p* < 0.01, *** *p* < 0.001. (**A**) Expression of FTL. The samples included 11 healthy donors, 3 providing cord blood and 8 bone marrow aspirates from adult donors, and 18 AML patients. The plot is the result of 42 and 54 measurements for healthy donors and patients, respectively. (**B**) Expression of FTH1. The samples included 11 healthy donors, 3 providing cord blood and 8 bone marrow aspirates from adult donors, and 24 AML patients. The plot is the result of 39 and 74 measurements for healthy donors and patients, respectively. (**C**) Expression of HMOX1. The HMOX1 transcript of each leukemic sample (grey bars) was amplified in triplicate at least, and its abundance is plotted as means and SD. CB (white bars) indicates cord blood progenitors. (**D**) Ratio of HMOX1/HMOX2 mRNA. HMOX2 mRNA was measured as HMOX1 in (**C**) and the pooled HMOX1/HMOX2 ratios were represented in the forest plot. This plot is the result of 9 and 33 measurements for healthy donors and patients, respectively.

**Figure 4 ijms-24-14307-f004:**
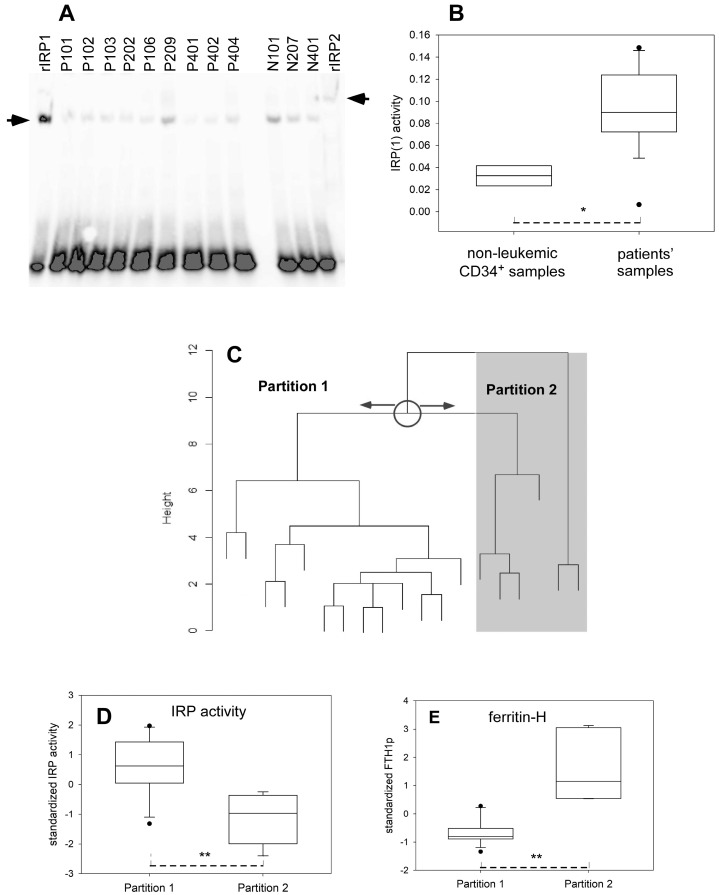
Activity of the iron regulatory proteins and partitioning of the AML blasts. (**A**) Representative blot of RNA electrophoretic shift assays probing IRE-binding. The blot includes a series of leukemic samples labeled P- for more than 20% CD34^+^ blasts, N- for less, and with recombinant IRP1 (rIRP1) and IRP2 used as migration markers. The intensity of the free probe (bottom of blot) indicates that the signal was saturated, hence, that the assays contained an excess of IRE needed to reveal all available activity (retarded band on top) present in the samples. (**B**) Forest plot of IRP activity comparing healthy progenitors and blasts. This plot is the result of 4 and 41 measurements for healthy donors and patients, respectively. * *p* < 0.05. (**C**) Dendrogram of the samples as partitioned with the *diana* algorithm. Divisive hierarchical clustering of the dataset including 19 samples and 6 variables is shown, but different algorithms produced qualitatively similar results (see text). (**D**) Forest plot of the standardized IRP activity variable (value divided by the root-mean-square) comparing the samples partitioned in (**C**). (**E**) Forest plot of the standardized ferritin H protein variable comparing the samples partitioned in (**C**). ** *p* < 0.01.

## Data Availability

All data are included in the main manuscript and the Appendix A. Additional non-reported data may be available from the corresponding author upon request.

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
