# Peer review of "Molecular Insight into Iron Homeostasis of Acute Myeloid Leukemia Blasts"

_ijms, 2023, doi:10.3390/ijms241814307_

Round 1

Reviewer 1 Report

Molecular insight into iron homeostasis of acute myeloid leukemia blasts

Emmanuel Pourcelot et al (2023)

Summary of Paper: 

Acute myeloid leukemia (AML) is a tragic disease with a low 5-year survival rate. Due to the heterogeneous genetic and metabolic characteristics exhibited by leukemia cells, understanding their outgrowth is crucial for the development of new therapeutics. Iron plays a vital role in various biological functions; therefore, it has been extensively studied in various diseases such as cancer and AML. In this study, Emmanuel et al. investigated the proliferation status of both AML blasts and hematological CD34+ progenitors, with or without iron, to comprehend the contribution of iron. While freshly isolated CD34+ progenitors remain mostly quiescent, the fleshly isolated leukemic samples have a dividing population. Supporting this observation, mRNA levels of RRM2 were found to be increased in AML patients. Upon stimulation by adding transferrin to the media, proliferation of both progenitors and leukemia-initiating cells occurs, although the authors noted a high population of cell death. To analyze iron dependence, the authors measured cell proliferation in response to increasing concentrations of transferrin, finding that cell proliferation plateaued at a concentration of 100 nM. Consequently, the expression of the transferrin receptor decreased. Furthermore, the iron responsivity of AML was lower than that of progenitors from healthy controls, which was consistent with the high TFRC expression in AML patients. Next, the authors aimed to estimate intracellular iron storage by measuring the expression of FTL and FTH1; they observed an increase in the expression of both genes in AML patients. The expression of FTL and FTH1 is regulated by the activity of IRPs, and the authors found that AML patients exhibited high IRP1 activity. Subsequently, using data about transferrin receptor, ferritin protein and mRNA levels, and IRP activity for partitioning, the authors categorized the AML patient samples into two groups. This categorization revealed significant differences in IRP1 activity and FTH1 expression between the groups. 

This manuscript showed dysregulation of iron homeostasis in AML disease and they also provided subgroup in AML patients. Whilst the paper is interesting, some of issue to be solved to solidify their results.

Comments/Experimental suggestions:

Figure2;

1.     overall, the concentration range of transferrin varies widely between experiments.

2.     The content of line 206 (If AML blast~) is already known. It needs to be corrected.

Supplement Figure 2

Increase of PDK1 and LDHA expression is already known. Please cite papers.

Reviewer 2 Report

The idea presented in this work is potentially interesting. Classifying AML cells based on their iron metabolism may help clinicians better stratify patients and improve therapy. An example might be the chance of being able to identify patients potentially susceptible to ferroptosis.

Having said that, the results shown in the manuscript look rather preliminary and do not fully support the conclusions. Authors define leukemic cells more prone to proliferation, but they do not proliferate more than control cells when considered as a whole. Indeed, it seems that leukemic cells are more susceptible to cell death.

Based on the variability observed in testing RRM2 and TFRC expression in leukemic cells (not surprising, in my opinion, since they are primary cells), the authors refer to proliferation-prone samples (patient), but ultimately they do not associate this predicted fast proliferating phenotype to an iron condition, nor authors investigate the cell oxidative state. In addition, iron levels are missing. For example: do cells with high TFRC expression correlate with cells with high or low PDK1 or LDH1? Which group of cells is more proliferating/resistant to drugs..

Proliferation studies and drug testing comparing AML cells with different iron signatures coming from different patients are needed in my opinion to fulfill the message.

Moreover, is there any correlation between iron signature and patient outcome?

Minor concerns:

Figure legends could be condensed as well as the discussion

It’s not clear how gene expression values are plotted in the graphs. Multiple measurements made on an individual primary sample should be used to generate a mean value referred to that sample. Then, mean values should be used to graph results and make statistical comparisons.

Figure 2. A discrepancy emerges between the text and the figure legend in terms of cell proliferation. In the text the authors refer to leukemic cells as non-proliferating, while in the figure legend the increase of proliferation after iron addition respect to no iron is reported as statistically significant.

Figure S2. The reported differences in PDK1 and LDH1 gene expression between controls and AML cells is not sufficient in my opinion to ultimately refer for a shift in cellular metabolism. Moreover, a discrepancy emerges again between the text and figure legend in terms of statistically significance of the results.

Figure 4. Assuming that samples initiating with P are leukemic cells (please, add a reference in the legend), it’s not clear why authors refer to a higher IRE-binding activity in leukemic cells than in controls (starting with N?). Moreover, IRP1 and IRP2 are not predicted to co-migrate in RNA shift assay of human samples?

Round 2

Reviewer 2 Report

The authors introduced enough discussion points in their response to address all of my minor comments and even some of the major ones. I remain of the opinion that the work, to be complete, should be correlated with in vitro studies on proliferation and/or drug sensitivity.

I have no further comments as a reviewer on the experiments presented in the manuscript.

Round 3

Reviewer 2 Report

I have no additional comments.